# Smartphone use behavior and quality of life: What is the role of awareness?

**Alon Sela** [ID]**[1]\*, Noam Rozenboim[2], Hila Chalutz Ben-Gal[3]**

**1** Department of Industrial Engineering, Ariel University, Ariel, Israel, **2** Department of Industrial Engineering, Tel Aviv University, Tel Aviv, Israel, **3** Department of Industrial Engineering and Management, Afeka Tel Aviv Academic College of Engineering, Tel Aviv-Yafo, Israel

\* alonse@ariel.ac.il

**Data Availability Statement:** The data and code are available in the Github repository of the article (https://github.com/alonsela2015/Smartphones_

## Abstract

How does smartphone use behavior affect quality of life factors? The following work suggests new insights into smartphone use behavior, mainly regarding two contradicting smartphone modes of use that affect quality of life in opposite ways. The Aware smartphone mode of use reflects an active lifestyle, while the Unaware mode of use reflects the use of the smartphone in conjunction with other activities. Using data from 215 individuals who reported their quality of life and smartphone use habits, we show that high levels of smartphone use in the Unaware mode of use have a significant negative effect on the quality of life. However, the results show a mild positive effect when the individual uses the smartphone in an aware mode of use. We identify three latent factors within the quality-of-life construct and measure the effect of the different smartphone modes of use on these quality-of-life factors. We find that (i) The functioning latent factor, which is an individual's ability to function well in his or her daily life, is not affected by smartphone use behavior. In contrast, (ii) the competence latent factor, which is a lack of negative emotions or pain, and (iii) the positive feelings latent factor both show a clear effect with the smartphone Unaware mode of use. This implies that the unaware use of smartphones, which is its use in conjunction with other activities or late at night, can be related to lower levels of quality of life. Since smartphones currently serve as an interface between the self and the cyber space, as well as an interface between the self and other individuals online, these results need to be considered for social wellbeing in relation to digital human behavior, smartphone addiction and a healthy mode of use.

## Introduction

Smartphones are everywhere. As of April 2020, approximately 3.8 billion people in the top fifty most-populated countries used smartphones [1]. Smartphones are nowadays an entry point to the Internet, a portable computer that speeds up the transfer of knowledge, enhance e-commerce, and as such are important for economic growth. This is also why their use rates is a part of the Global Competitive Index [2]. While some research suggests that smartphone use can be categorized as dangerous (e.g., using a device while driving), inappropriate (e.g., using a

QualityOfLife). Github address is also found in the reference section.

**Funding:** This research was partially supported by The Koret Fund of Digital Living 2030. This research was partially supported by Ariel Cyber Innovation Center. The funders had no role in study design, data collection and analysis, decision to publish, or preparation of the manuscript.

**Competing interests:** The authors have declared that no competing interests exist.

device in a cinema or class), or simply excessive [3, 4], smartphones are currently rooted in modern society to such degree that prohibiting or even reducing their use is perceived as a punishment by both adults and younger adolescents.

How do various patterns of smartphone use relate to quality of life (QOL) factors? This multidimensional question contains several research branches. The first emphasizes physical and mental health well-being as factors that are influenced by smartphone use [5–11]. An alternative view focuses on the challenges that smartphones pose for work and family balance [12, 13], as well as stress levels among individuals [14–23]. It seems that the definitions of QOL itself are broad and that there are many methodologies that aim to study this complex but crucial aspect of our lives. QOL research spans the fields of psychology [24], philosophy [25], social psychology [26] and economics [27]. Although the QOL literature uses different terms to define the term QOL, it is broadly accepted that QOL can be associated with having a good life [28], e.g., one's degree of success in reaching a level of satisfaction within the constraints of the resources that one possesses, or factors such as feelings of joy, pleasure, and life satisfaction, which are important constructs of QOL [24, 29]. In recent years, the academic discussion about QOL has also included concepts such as work-life balance [30], which complement the previous QOL measures. Also, a recent study [27] has found that QOL measurement for young adults should also include the Economic Factors.

The literature that examines the complex interaction between smartphone use and QOL is expansive, but the empirical evidence regarding the relationship between smartphone use patterns and its consequences for quality of life are mixed.

In this work, we follow recent calls [31, 32] and study QOL factors as a complex phenomenon related to smartphone use behavior. Our goal is to deepen our understanding of the different effects of smartphone use on QOL to better understand the contradictory findings. To achieve this goal, we assume that it is not the smartphone itself but rather the individual's mental state while interacting with the smartphone that affects one's QOL and thus, that it is this type of interaction that should be examined to study the relationship between smartphone use and QOL. Following this line of study, we identify two modes of smartphone user behavior, which we define as the *Aware* and *Unaware* smartphone use modes.

These mental states are defined by in order to settle some contradictory findings in terms of QOL and states of awareness. For example [31], showed that deep levels of awareness can be associated with higher levels of QOL, while [33] claimed, conversely, that low levels of awareness lead to varying levels of QOL. Additionally, it is well known that when individuals are in low-awareness modes, their ability to accomplish tasks may suffer [34]; it is also known that high-awareness modes improve one's performance, for example, in medical situations [35]. These contradictory effects might impact QOL. We thus measure the relationship between the effect of smartphone use on QOL in relation with an aware and unaware modes of smartphone operation.

To measure QOL, we use the QOL questionnaire developed by [36]. This is a multidimensional tool that has already been shown to be a valid evaluation questionnaire for measuring QOL. In this tool, to construct the final level of QOL, the questionnaire measures three different subcomponents that together form the multidimensional QOL estimates. These subcomponents are named *Competence*, *Functioning* and *Positive Feeling* (see Table 2 for the question construction of these subcomponents). Generally, competence is defined as the presence (or when normalizing, the lack of) negative feelings such as pain, lack of control, fear, or anxiety. Functioning is defined as feelings of strength, independence, and motivation. Positive feeling is defined as the presence of hope, pleasure, satisfaction and, to a large degree, the ability to maintain good relationships with others. We test the relations between these five latent variables (i.e., the three QOL factors and the two smartphone use modes) using structural equation

modeling (SEM) techniques. In addition, as would be explained, we use conventional statistical methods such as analysis of variance and dimensionality reduction methods that are borrowed from the data science discipline to optimize the separation into these subcomponents.

Our empirical results support our hypotheses that high levels of *unaware* smartphone use are harmful to QOL, while high levels of *aware* smartphone use are not found to significantly affect QOL parameters. Thus, our findings suggest that it is not simply the magnitude of smartphone use that needs to be controlled and limited within the effort to balance the negative effect of excessive and harmful smartphone effect to QOL but rather the mode of operation when one is using a smartphone and particularly, in particular, the degree to which coactivations with other tasks are performed during this use. As will be further shown, these coactivations seem to be of high importance regarding the harmful aspects of excessive smartphone use that negatively affect QOL.

## Methods

### Data

We collected 215 questionnaires through Amazon's Mechanical Turk crowdsourcing marketplace, which included questions related to both smartphone use habits and quality of life factors. The Respondents completed the survey task in an average of 5.7 minutes and were paid 0.5 USD per task. The sample size was computed by an initial wave of 50 respondents, (see section 4 in S1 File for sample size related details), which helped estimating the required final sample size. The questionnaire was constructed from two separate sub-clusters of questions, (which can be clearly visually observed in Fig 1). The first is of the QOL questions, and the second of the usability. It is important to note that the QOL questionnaire is a known questionnaire with validated reliability [36]. Regarding the reliability of the usability items (see section 3 in S1 File for details), while the Unaware cluster resulted in a good reliability (Cronbach's Alpha = 0.761, McDonald's Omega = 0.774), the Aware cluster reliability was weaker (Cronbach's Alpha = 0.592, McDonald's Omega = 0.594). Nevertheless, we kept this latent variable and did not combine the Aware and Unaware variables to one single usability factor, because the coefficient of the Unaware constructs where influencing QOL negatively, while these of the Aware constructs effected QOL positively.

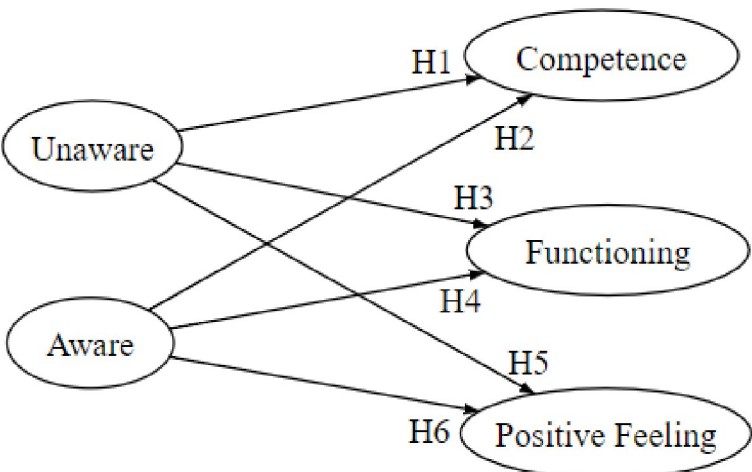

**Fig 1. Proposed theoretical model and hypotheses.**

We also performed a Confirmatory Factor Analysis (CFA) to access the reliability of the questionnaire. The full CFA results are presented in the section 2 in S1 File and support the questionnaire validity as the CFI = 0.913 and TLI = 0.928 (reasonable); the RMSEA = 0.049 – (<0.05 is good) and SRMR = 0.061 (values less than 0.08 are considered as good). These criteria, together with the cross-validation results suggested that although the aware construct is weak, SEM could be safely applied to our data and of an accepted reliability of the questionnaire.

All data, including the questionnaires and the statistical analysis in R and Python, can be found and downloaded from the GitHub repository [37] related to this work. The gender distribution favored male respondents, with 38% female and 62% male respondents. Participants' level of education averaged 15.6 years (SD = 3.15), and the participants reported having owned their smartphone for an average of 5.48 years (SD = 4.03) at the time of data collection. Additionally, the participants reported that they had installed an average of 30 applications on their smartphones. Interestingly, only 13.6% of the female respondents reported using the device "10 times each hour," compared to 64% of the male respondents. In addition, 13% of the female respondents reported using their smartphone "almost always" while performing another activity, such as driving, cooking or watching television, compared to 60.7% of the male respondents. Additionally, 22.7% of the female respondents reported using their phone more than 4 hours on average per day, compared to 71.4% of the male respondents.

The questionnaire included a shorter version of the QOL factor questions, which were validated against the full QOL questionnaire [36]. The questionnaire also included a set of 12 questions that inspected major smartphone usability habits from three different aspects of use. The usability questions inspected two types of smartphone use behaviors. The first type can be broadly generalized as the intensity of use, i.e., the number and type of applications that a user utilizes while using the phone. The second type consisted of the user's mental mode of smartphone use, and the related questions inspected the use of smartphones in parallel with other user activities. These questions were based on what we believe could be an important moderating factor in the usability-QOL interaction. The third type of question included the user's report on his/her beliefs of his/her addictive use behavior, as well as the degree to which he/she believes that a smartphone is an indispensable part of one's social and professional life.

From the initial 215 surveys that were collected, 4 surveys were removed due to the poor quality of data within. In the remaining 211 surveys, with respect to the respondents' occupations, 74% reported working full-time jobs, and the rest reported working either part-time or not working at all. A total of 98% reported having one single phone, and the rest reported having two or even three phones. Last, all the participants reported that they had owned their phone for at least six months. This question was important to ensure that the participants had owned a phone long enough for its use to have impacted their lives.

## Ethics

The experiment was approved by the Ethical Committee of Tel Aviv university IRB for research proposal no. 0002152–2, given to Dr. Hila Ben-Gal. The participant consent informed was written and they were informed by the statement "*We are conducting an academic survey about smartphone usage habits and other aspects of life. We need to understand your opinion about the questions in the survey. Press the link below if you agree to participate in the survey*". Only users that actively pressed the link after reading this statement could participate in the survey. After completing the survey, each responder was paid 0.5 USD from the Amazon Mechanical Turk tasks system.

## Quality of life measurement

We based the QOL measurement on the Multidimensional Quality of Life Inventory (MQOL) questionnaire [36]. This scale includes twenty-two items that represent fifteen different domains of QOL: functioning in the family, physical health, living conditions, sexuality, body image, cognitive functioning, work and profession, social functioning, presence of positive emotions, presence of negative emotions, meaningfulness of life, confusion and bewilderment, ability to cope and expression of stress. A higher score represents better QOL results. Additionally, sections of the questionnaire include measures related to psychographic details, stress levels and demographic and control variables.

## Hypotheses and latent variables construction

We started by analyzing the data to determine how to best separate the variables into their latent subcomponents, i.e., latent variables for both QOL and smartphone usability. These latent variables were then employed in our model and in the questionnaires. We followed conventional data mining and machine learning methodologies and borrowed methods from disciplines presented in [38–40]. According to this analysis path, the study first explores the data through known statistical tools and then build a learning model to better explain the results that were emerged from the data analysis stage. Therefore, we first conducted a preliminary dimensionality reduction procedure, where each distinct question in the questionnaire was considered as a dimension, in order to find the most informative dimensions in the questionnaire. Note that in such an analysis, the higher dimensions are conceptually equivalent to Latent Variables that are constructed of numerous questions vectors.

Following this preliminary analysis, we found that it is best to separate the QOL questions into three main subcomponents (i.e., the latent variables) and the smartphone usage questions into two latent variables. The full description of the procedure that was used to find the number of latent variables as well as how to determine the questions that appear in each latent variable are now presented in Figs 2 and 3, Tables 1 and 2 in the results section below.

In accordance with the QOL subcomponents developed by [36], we named the three latent QOL variables—'Competence,' 'Functioning,' and 'Positive feeling' as in the original QOL scale. To better grasp the items' meaning, one can look at Table 2, where the correlation coefficients

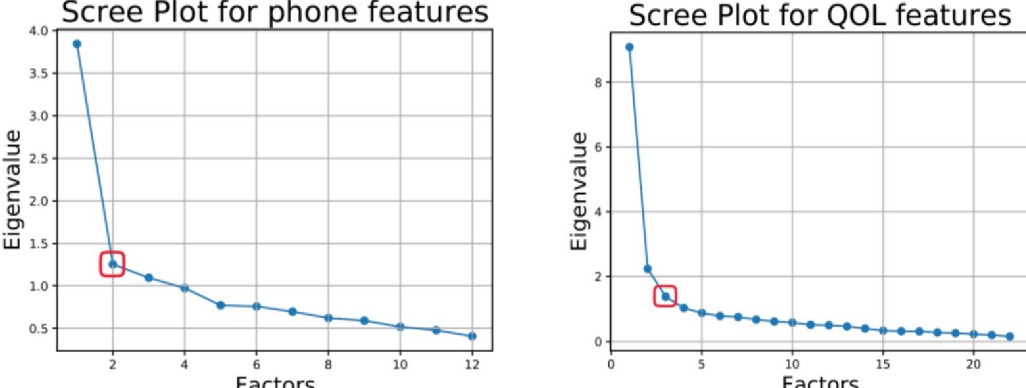

**Fig 2. The number of latent variables as determined by the elbow method.** These plots show the number of latent variables associated with the smartphone use modes (left) and quality of life (right). The x-axis is the number of latent variables, and the y-axis is the additional variance that can be explained for each additional latent variable. Note that for the phone use factor, the elbow method determined 2 latent factors as an effective separation, while for the QOL factor, it determined 3 latent factors.

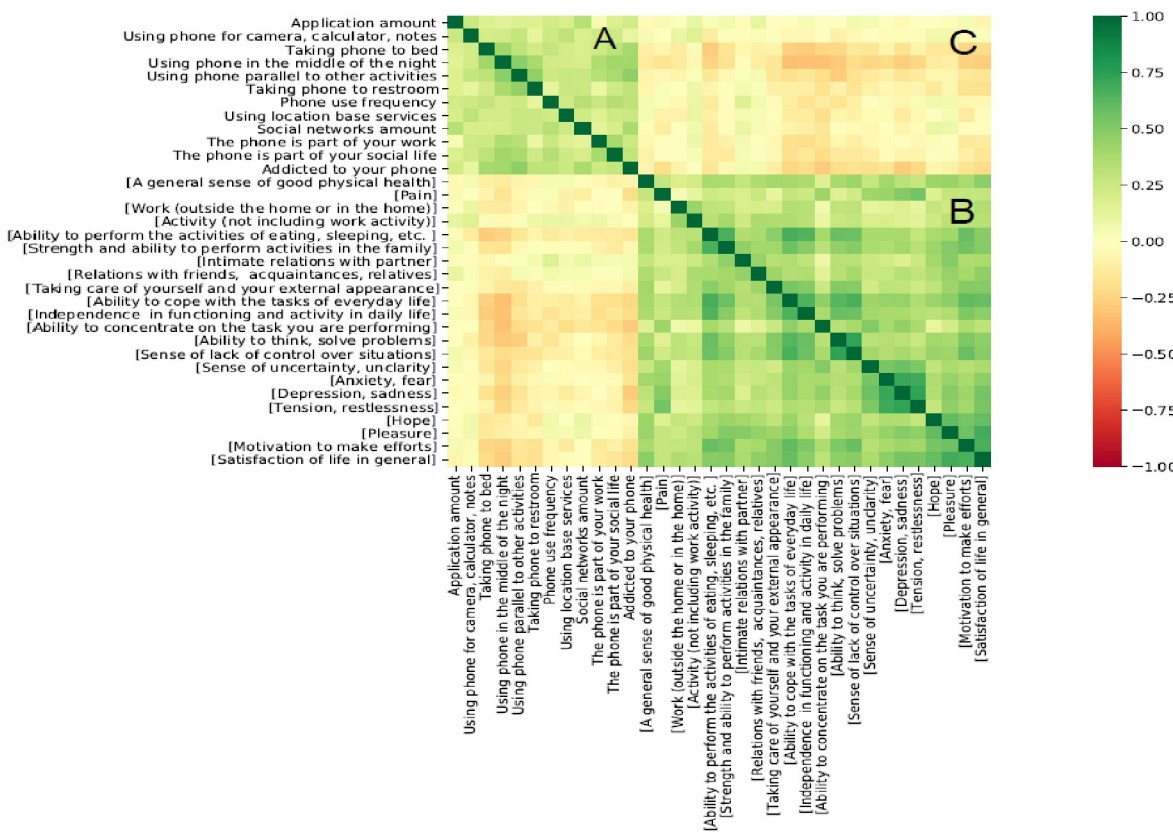

**Fig 3. Correlation matrix of questionnaire items.** Green indicates a strong correlation, and red/yellow indicates a weaker correlation. Smartphone use behavior (A) and quality of life items (B) show a good separation, which forms two main clusters of items, one related to QOL and the other to smartphone use behavior. (C) shows weaker relationship in comparison to each individual sub-component.

between the items and the three QOL latent variables are presented. Generally, positive feelings is a cluster where people experience feelings such as 'hope', 'pleasure', 'satisfaction', 'good relations with others' and 'good health'. Competence includes feelings such as 'sense of control over life's situations', 'clarity', 'lack of anxiety', 'lack of depression' or 'lack of tension'. Functioning includes feeling of 'active lifestyle', 'functioning in work', and feelings of strength in performing basic daily activities, such as sleeping, eating, and overall, a good ability to easily perform basic daily tasks. Similarly, we named the two-smartphone usability latent variables *'aware'* and *'unaware'* use behavior modes as these names represents the common factor of the questions that are highly correlated with each latent variable.

The full description of the method used to separate the latent variables is further described in the results section and in Figs 2 and 3, and Tables 1 and 2.

To conclude, the descriptions of each latent variable are provided below.

**Unaware smartphone use.**   Unaware smartphone use represents smartphone use that is simultaneously performed with other activities that demand attention, such as cooking, driving, and watching television. Similarly, an important component of unaware use is the use of a smartphone in the middle of the night, where one's level of attention is usually lower due to tiredness; thus, the activities performed while using the smartphone may be associated with its use while the individual is expressing lower levels of concentration.

**Aware smartphone use.**   This smartphone use is characterized by actively utilizing the numerus functionalities of the smartphone device, including its built-in features, for example,

**Table 1. Correlation coefficient between each item and the two smartphone use latent variables.**

| Variable | Smartphone Use Items | Aware | Unaware |
|---|---|---|---|
| x1 | How many applications that are being used do you have on your smartphone/cellular phone? | -0.18 | 0.64 |
| x2 | Do you use your smartphone/cellular phone for activities that can be performed with other means, for example, camera, calculator, notes, reading books, making payments? | 0.1 | 0.4 |
| x3 | Do you go to bed with the smartphone/cellular phone at your side? | 0.44 | 0.12 |
| x4 | Do you use your smartphone/cellular phone in the middle of the night? | 0.85 | -0.2 |
| x5 | Do you use your smartphone/cellular phone in parallel with other activities, such as driving, cooking, watching TV? | 0.73 | -0.02 |
| x6 | Do you take your smartphone/cellular phone with you to the restroom? | 0.45 | 0.05 |
| x7 | Frequency of using your smartphone/cellular phone: on average, how many times do you use it each hour during the day? | 0.11 | 0.4 |
| x8 | Do you use the location-based services of the smartphone/cellular phone? | 0.12 | 0.4 |
| x9 | How many social networks (e.g., Facebook, Twitter) are downloaded on your smartphone/cellular phone? | 0.03 | 0.56 |
| x10 | Do you consider your smartphone/cellular phone to be an indispensable tool for your work? | 0.52 | -0.06 |
| x11 | Do you consider your smartphone/cellular phone to be an indispensable tool for your social life? | 0.52 | 0.08 |
| x12 | Do you consider yourself to be addicted to the use of your smartphone/cellular phone? | 0.47 | 0.17 |

using its camera and calculator or the intensive use of email, calendar, notetaking, etc. These smartphone functionalities might reflect an active and busy lifestyle. Note that the two use modes do not by nature contradict each other; theoretically, a person might show high levels of aware use together with high levels of unaware use at the same time.

**Competence.**   The competence latent variable refers to an individual's perception of the degree to which they believe they can cope with life. This latent variable is also associated with QOL items that reflect the existence of negative feelings, such as pain, a sense of lack of control of one's life, a sense of uncertainty, and feelings of anxiety, fear, depression, sadness, tension, and restlessness.

Following the definitions above, we hypothesize the relationship between *smartphone use* and the *competence* latent variable of QOL as follows:

H1.   *Higher levels of unaware smartphone use are associated with lower levels of competence.*

H2.   *Higher levels of aware smartphone use are associated with higher levels of competence.*

**Functioning.**   The functioning latent variable in QOL [36] examines whether a person functions well in his/her basic daily routine, both at home and at work. This latent variable refers to numerus attributes of regular daily and life functioning dimensions, such as one's healthy physical functioning, a good level of functioning within the family, normal cognitive abilities, e.g., the ability to concentrate and the ability to solve problems, and one's ability to keep healthy relationships with family and friends. It also includes QOL items such as the ability to sustain a job, the performance of regular physical exercise, the ability to perform basic life activities such as eating, sleeping, or taking care of yourself, and in general, the feeling of being able to cope with the basic requirements of one's daily routine of life.

We believe that being capable of using numerus and technically complex applications on a smartphone, such as texting, using reminders and notes, using GPS-based location services or

**Table 2. Correlation coefficient between each item and the three QOL latent variables.**

| Variable | Quality-of-life Item | Functioning | Competence | Positive Feeling |
|----------|---------------------|-------------|------------|------------------|
| Y1 | A general sense of good physical health | 0.17 | 0.01 | 0.38 |
| Y2 | Lack of pain | -0.04 | 0.61 | 0.02 |
| Y3 | Work (outside the home or in the home, including household work) | 0.43 | -0.15 | 0.14 |
| Y4 | Activity (outside the home or inside the home, not including work activity) | 0.34 | -0.1 | 0.25 |
| Y5 | Strength and ability to perform the activities of eating, sleeping, etc. | 0.8 | 0.04 | -0.02 |
| Y6 | Strength and ability to perform activities within the family (as a partner, parent, sibling, son/daughter) | 0.41 | 0.06 | 0.32 |
| Y7 | Intimate relations with your partner | -0.01 | -0.05 | 0.61 |
| Y8 | Relations with your friends, acquaintances, relatives | 0.09 | 0.04 | 0.45 |
| Y9 | [Taking care of yourself and your external appearance | 0.49 | -0.06 | 0.27 |
| Y10 | Strength and ability to cope with the tasks of your everyday life | 0.81 | 0.12 | -0.03 |
| Y11 | Independence in functioning and activity in your daily life | 0.78 | 0.07 | -0.04 |
| Y12 | Sense of control over situations, feeling that you can determine what happens | 0.38 | 0.53 | -0.34 |
| Y13 | Ability to concentrate on the task you are performing | 0.76 | -0.04 | 0 |
| Y14 | Ability to think, solve problems | 0.76 | -0.04 | 0 |
| Y15 | Sense of certainty, clarity | 0.1 | 0.65 | -0.04 |
| Y16 | Anxiety, fear (lack of) | -0.06 | 0.74 | 0.15 |
| Y17 | Depression, sadness (lack of) | -0.12 | 0.84 | 0.17 |
| Y18 | Tension, restlessness (lack of) | -0.22 | 0.8 | 0.24 |
| Y19 | Hope | 0.09 | 0.02 | 0.63 |
| Y20 | Pleasure | -0.07 | 0.13 | 0.81 |
| Y21 | Motivation to make efforts and continue doing things | 0.48 | 0.04 | 0.33 |
| Y22 | Satisfaction with life in general | 0.2 | 0.18 | 0.56 |

using social media, should be positively correlated with the functioning latent variable of QOL. Technical competence in using complex applications should reflect (to some degree) basic cognitive functioning, and these basic cognitive functioning are expected to improve one's ability to perform basic daily tasks. While the use of numerus complex applications should reflect the normal functioning of basic cognitive skills and support a better QOL, we also suspect the existence of a contradicting effect in which an overuse of numerus complex applications on one's smartphone might be a sign of cognitive overload. Thus, due to this effect, such overuse is expected to be correlated with lower levels of the functioning latent variable and lower levels of QOL. Therefore, we hypothesize as follows:

H3. *Higher levels of unaware smartphone use are associated with lower levels of the functioning latent variable of QOL.*

H4. *Higher levels of aware smartphone use are associated with higher levels of the functioning latent variable of QOL.*

**Positive feeling.** The positive feeling latent variable in QOL measurement [36] examines the degree to which an individual experiences positive emotions. It is to some degree the opposite of the competence latent variable, in which one feels helpless, depressed and incompetent about one's life. It also differs from the functioning variable, which is more external and measures one's functionality and instead of one's emotions. That is, an individual can be fully functioning but also depressed (high levels of functioning and low levels of competence). Similarly, we expect it to be less likely that one would experience high levels of positive feeling and high levels of competence at the same time. The positive feeling variable is characterized by QOL

items that refer to the regular experience of positive health, a general sense of hope and the ability to feel pleasure. It also includes questions related to an ability to maintain healthy relationships with one's associates, friends and intimate partners.

We hypothesize that intensive use of a smartphone for communication purposes helps with one's everyday communication with friends and family and thus improves one's overall QOL satisfaction and level of positive feelings. We also believe that intensive unaware smartphone use, among other things, might determine the degree to which one is "using a smartphone in the middle of the night,", thus, might lead to sleep deprivation and reflect signs of addictive smartphone behavior. These could lead to lower levels of the positive feeling latent variable and to overall lower levels of QOL. Thus, we define two additional hypotheses related to the relationship between smartphone use and the QOL positive feeling latent variable. These are defined as follows:

H5. *Higher levels of unaware smartphone use are associated with lower levels of the positive feeling latent variable of QOL.*

H6. *Higher levels of aware smartphone use are associated with higher levels of the positive feeling latent variable of QOL.*

The six hypotheses above are summarized in the illustration below. These hypotheses will be examined and evaluated by utilizing the structural equation modeling method (SEM) [41], which is implemented through the Lavaan package in R [41].

## Results

### Defining latent variables by dimensionality reduction

In order to define the best constructs of the different sub components, one common method that is used in data science to determine latent variables and construct higher-level hierarchical classes from individual questions is the use of *dimensionality reduction.* According to this approach, the elbow method [38] is commonly used as a practiced heuristic that helps to determine, based on the data alone, the best number of latent variables that represent the exploratory factor analysis [42, 43].

Following to this method, we first used the elbow method on an x-y plot, where the number of latent variables (x-axis) was plotted against the corresponding eigenvalues (y-axis). These eigenvalues represent the additional variance that can be explained from the total variance of different methods of separation into lower dimensions. The plot usually has a large negative derivative at the first eigenvalues, which then quickly decays as more dimensions (eigenvalues) are introduced into the model. This process creates a plot with an elbow shape, (see Fig 2) with the breakpoint of the slope, i.e., the elbow, representing the optimal number of latent variables that best fit the model. We used this method to choose the best number of latent variables for the smartphone use dimension, as well as for the QOL variable. We chose to use the elbow method mainly because of its good results, common usage and simplicity [44].

The best number of latent variables that separates each of the two major factors, i.e., smartphone use and QOL, is presented in Fig 2. The left image presents the optimal number of latent variables for the phone usability factor. According to this method, the optimal number of subcomponents for smartphone usability is 2 components (left image). For the QOL factor, the optimal number of components is 3 latent variables (right image).

## Inner correlations of smartphone use and QOL

To understand and validate the interaction between smartphone use behavior and QOL factors, we also analyzed a correlation matrix of the entire data set. Fig 3 below presents three distinct zones in the full correlation matrix. The lighter (yellow) color represents a lower correlation, while the green color represents a higher correlation. Zone A shows the inner correlation within the smartphone use behavior questions. As illustrated by the green square in A, users who scored high on some dimensions of smartphone use also tended to score high on the others. Similarly, zone B presents the inner correlation matrix within the QOL factors. Similarly, inner validity is well observed in zone B. Finally, zone C represents the correlation matrix between smartphone use behavior and QOL factors. However, as illustrated in the image, these correlations are not as strong as the inner correlations within each topic (i.e., zone A and zone B), which, due to their relatively stronger inner correlation, change the entire image color to a greenish color; the QOL usability matrix in zone C seems to be weaker in comparison.

## Determining the questions that construct the latent variables

In the previous section, we determined the best number of latent variables for QOL and smartphone usability. Next, we needed to define what questions should be assigned to each latent variable. Deciding on the latent variables required a preliminary analysis that determined which observed variables (questions) belong to each latent variable. We performed this analysis using the observed variable loading method [45]. According to this method, the loading factors of each question represent the relationship of each question to the underlying latent variable [46, 47]. The loading value is the correlation coefficient between the observed variable and the latent variable. The higher the loading factor is for each question and latent variable, the better the question represents the latent variable. According to this method, loading value of under 0.3 means very weak correlations that should be ignored. This analysis and the item loading results are presented in Table 1 for smartphone use and in Table 2 for QOL. Note that the green-colored cell signifies the binary decision regarding the best allocation of each question to the most suitable latent variable.

## Model evaluation

We used the Structural Equation Modeling (SEM) method to examine the complex sets of interactions between the use behavior variables and the quality-of-life variables. We tested our model fit using the following criteria [48]: comparative fit index; CFI = 0.913; Tucker-Lewis index; TLI = 0.928; root mean square error of approximation; RMSEA = 0.049; and standardized RMR SRMR = 0.06. These criteria suggested that SEM could be safely applied to our data.

Additionally, we performed an explanatory factor analysis to validate the fitness of the different latent variables. As seen in the CFA analysis (see section 2 in S1 File), we achieved excellent fit indices. While at first glance this excellent fit might seem unclear, or even suspicious, one needs to consider that the method used in this analysis determined the latent factors from the data itself through the dimensionality reduction method and not by making a predominant assumption about these latent variables. Because we first found the optimal number of latent variables and then allocated each question to the most adequate latent variable, our use of this method and the subsequent allocation of the questions to the latent variable resulted in an almost perfect CFA. Indeed, in many cases in the social sciences, the theory used leads the separation of the questions into different latent variables. We took the data science path and first analyzed the data. Then, through this initial analysis, we found new possible insights based on the data and then supported them statistically.

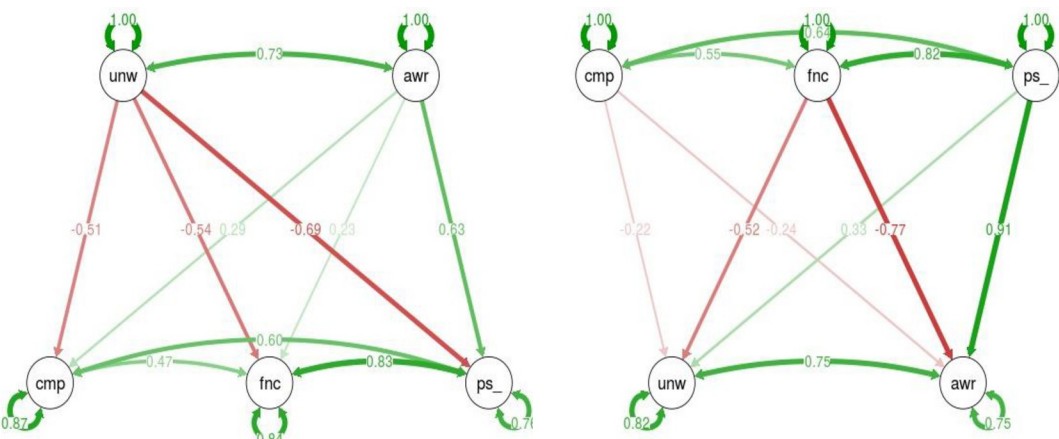

**Fig 4. SEM models.** Smartphone usability effects on QOL (left) and its inverse model, effects of QOL on usability (right). For the direct model (left), unaware use (unw) negatively effects all three QOL measures: competence (cmp), functioning (fnc) and positive feeling (ps_), while aware use (awr) positively effects positive feelings (ps_) but does not significantly affect the functioning or competence latent variables. For the inverse model (right), the functioning (fnc) QOL latent variable negatively effects both aware (awr) and unaware (unw) usability, while competence does not significantly affect either of the latent variables of aware or unaware use; furthermore, positive feelings (ps_) positively effects the aware component of QOL but not the unaware component.

## Structural equation modeling analysis

To evaluate our hypotheses, we used structural equation modeling (SEM) package in the Lavaan Package of R Software) [41]. Fig 4 shows the connections among the different latent variables in our model, with red arrows representing negative coefficients and green arrows representing positive coefficients. The values on the arched links are the normalized correlations between different latent variables of the same factor, while the values on the straight lines are the regression coefficients between the source and the target. In the first model, we analyzed the relations between the smartphone use modes (aware and unaware) and the three latent variables that form QOL (competence, positive feeling, and functioning). We measured both the direct model, in which we inspected the effects of smartphone usability on QOL in Fig 4 (left), and the inverse model, in which we inspected the inverse effects of OQL on usability in Fig 4 (right). Overall, the entire SEM model had good results, with a comparative fit index (CFI) = 0.934; a Tucker-Lewis index (TLI) = 0.928; a root means square error of approximation (RMSEA) = 0.004; and a standardized RMR (SRMR) = 0.06.

The direct SEM model found support for the relationships between the QOL factors and phone use. This model implies a strong connection between smartphone use and QOL; however, it could be that it is the QOL that affects smartphone use and not the opposite. Fig 4 (right) demonstrates the opposite direction model. High inverse estimates (with p-value > 0.05) can be found between functioning (fnc) and unaware (unw) with correlation of -0.77, and functioning (fnc) and aware (awr) with correlation of -0.52. Additionally, strong correlations are found between positive feelings (ps_) and aware (awr) with correlation of 0.91. The rest of the estimates are very low and therefore show an insignificant connection between the additional latent variables.

Our analysis yields a complex set of relationships between smartphone use and QOL. Nevertheless, if we aggregate the two images and only consider the supported hypotheses, as presented in Table 3, we can conclude that using a smartphone in the unaware mode is likely to negatively influence all three components of one's QOL, while having a strong positive feeling

**Table 3. Summary of the structural model estimates and P-values.**

| Regressions | Hypothesis | Estimates | Quality of Life | | | Hypothesis |
|---|---|---|---|---|---|---|
| | | | SE | $P(>|z|)$ | Std.all | |
| Competence—Unaware | H1 | -0.409 | 0.147 | **0.002** | -0.505 | Support |
| Competence—Aware | H2 | 0.485 | 0.325 | 0.106 | 0.290 | Reject |
| Functioning—Unaware | H3 | -0.322 | 0.111 | **0.001** | -0.538 | Support |
| Functioning—Aware | H4 | 0.289 | 0.231 | 0.123 | 0.234 | Reject |
| Positive Feel.—Unaware | H5 | -0.496 | 0.155 | **0.001** | -0.694 | Support |
| Positive Feeling—Aware | H6 | 0.929 | 0.366 | **0.014** | 0.629 | Support |
| N = 215 | | | | | | |

component about one's QOL is likely to positively affect the aware mode of use of one's smartphone.

## Deeper inspection of the unaware smartphone use

To further understand the relationship between the unaware smartphone mode of use and QOL, we further inspected the effect of the unaware latent variable of QOL on each question. Note that for each of the 7 questions that form the unaware latent variable, the respondents were asked to grade the degree to which they agreed with each statement about themselves on a scale ranging from 0 to 2. The person's unaware mean score, denoted as $-S_U$, can thus be computed as the mean of these 7 answers. We separated the population into 3 separate groups according in their average unaware score. We ignored the middle group and only inspected the users with high and low unaware scores, which were denoted as the $\vec{H}$ and $\vec{L}$ groups, respectively.

By separating these groups, we were able to directly look for significant differences in the levels of QOL and compare the QOL of users who had high levels of unaware smartphone use with those who had low levels. Furthermore, we were able to look at each QOL component directly between these groups. Fig 5 shows the distribution of QOL, as well as the 3 separate QOL components, between these two groups. The positive feeling subcomponent had the strongest level of separation between the $\vec{H}$ and $\vec{L}$ groups (t-test, p-value<0.02), while the functioning subfactor difference was not significant (t-test, p-value = 0.23). We can therefore conclude that when measured directly, users with a low level of unaware smartphone use (blue) are likely to experience higher levels of competence and experience stronger levels of positive feelings, while the effect on functioning, although in a similar (negative) direction, is less clear.

## Discussion

The ever-growing importance of technology and smartphones in modern society is clear (Park and Lee, 2012). Smartphones have been even more important during COVID-19 lockdowns and quarantines, which require long stay-at-home routine, while connecting remotely to work, friends and family. In this article, we attempted to deepen our understanding of the effect of smartphone use behavior on different QOL factors. Following some disagreement regarding technology use behavior and our belief that it is not the technology itself but rather the interaction between individuals and the technology that may affect humans, both to positive or to negative feelings, we focused on the interface between humans and technology. Thus, this study analyzed specific smartphone use patterns to better understand the modes in

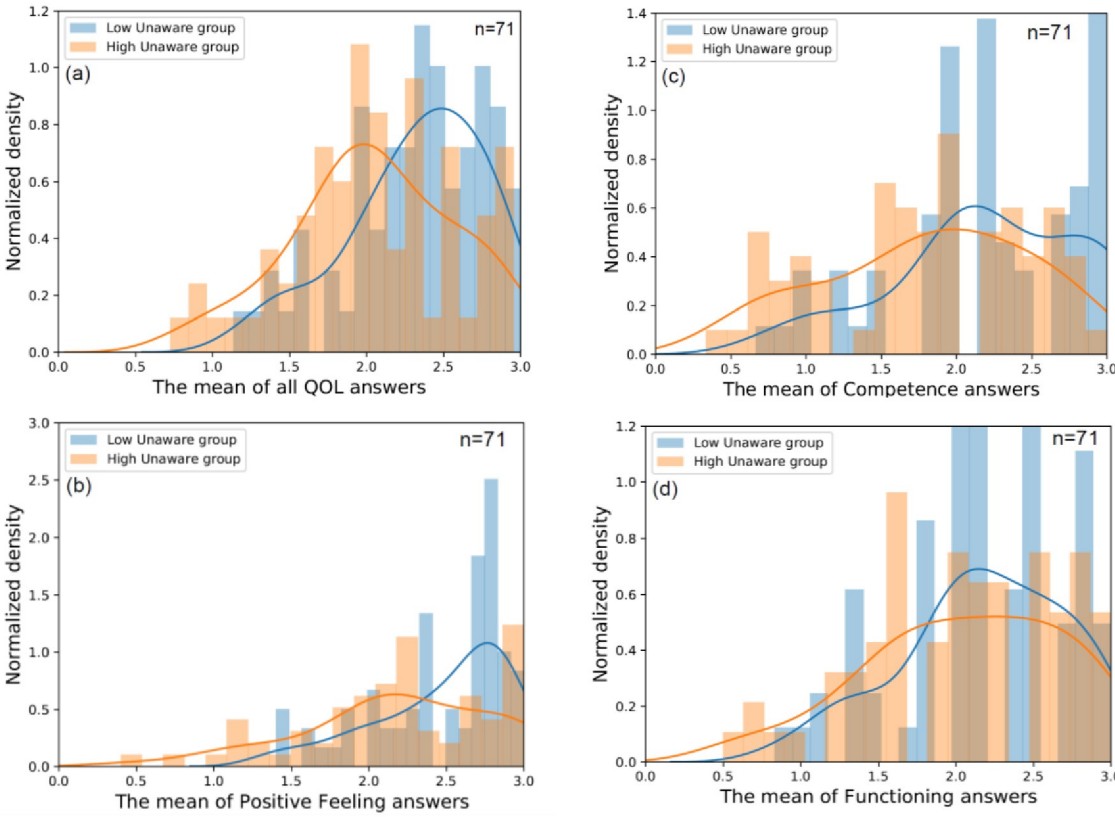

**Fig 5. Distribution of QOL scores for high unaware (orange) vs. low unaware smartphone use (blue).** In (a), we plot the histograms for the entire QOL component. In (b), we plot only the positive feeling subcomponent. In (c) we only plot the competence subcomponent. Finally, in (d), we plot only the functioning subcomponent. The exact questions that construct each QOL subcomponent can be seen in Table 2, while the questions that construct smartphone usability are found in Table 3.

which people use smartphones and how these modes contribute to QOL factors. We identified two main smartphone use behavior modes, namely, the aware and unaware modes of use. We also defined three QOL factors, named competence, functioning, and positive feeling.

We try to solve the disagreement between the claims that smartphones effects QOL positively, and those claiming its negative effect, by suggesting two opposite use effects on QOL. We found that while intensive use in the aware mode reflects an active lifestyle and contributes to positive levels of QOL, the unaware mode of use contributes to an opposite effect and results in reduced levels of QOL.

Looking at the relationships between the different subcomponents of smartphone use and the different subcomponents of QOL, we found that the strongest relationship is the negative effect between smartphone unaware use and the positive feeling subcomponent of QOL ($p = 0.001$). The strong effect of unaware use may be explained through the concept of *cognitive loading* [49, 50], which is a "mechanistic" theory that assumes that human cognitive resources are limited and thus, when these limited resources are distributed between too many tasks, performance is thereby reduced. This theory is also supported in the context of FMRI studies [51], in which cognitive loading can be directly observed and measured by analyzing changes in brain images. While FMRI methods can directly observe cognitive loading, the commonly used method to measure cognitive loading inspects the decrease in the performance of respondents when they are asked to simultaneously perform dual (or multiple) tasks

[52]. One should note that in our study, smartphone users were required to report whether they used their smartphone in parallel with other activities. We thus expected higher mental loadings [53] when smartphones were used in parallel with other activities. Cognitive loading is also known to increase stress levels [54], which results in aggression and negatively affects the ability of an individual to maintain good relationships with others. Relationships with others, on the other hand, are a major part of the positive feeling subcomponent in the QOL questionnaire (see Table 2). We thus see a direct link between the parallel performance of different tasks and smartphone use, which affects mental loadings, which contribute to stress and, in the long term, can harm one's good relationships with others and thereby reduce the positive feeling subcomponent of QOL.

Unaware smartphone use in conjunction with other activities may also be associated with addiction. For example, the examination of other types of technological addictions (e.g., internet addiction) has found that one well-validated predictive variable for internet addiction is a high level of attention deficit disorder (ADD) [55]. It is possible that people with some level of ADD might experience a higher tendency toward smartphone addictive use patterns, which in turn should reduce their level of the positive feelings QOL component. As demonstrated in Fig 4, the level of the positive feelings component does not reduce the results of the unaware QOL component, but high level of the unaware QOL component does reduce the positive feelings QOL component. This one-directional influence may suggest causality. In contrast, the relationship between aware smartphone use and the positive feelings QOL component is bidirectional, which indicates correlation and not causation. Thus, our results support the growing level of concern about the possible harmful mental implications of the excessive use of technological devices [56] while also focusing on the unaware mode of use.

The theory of planned behavior (TPB) [57] may partially explain our results from another perspective. According to the TPB, behavior is determined by an individual's intentions to perform the behavior. Intention is influenced by (i) attitude, (ii) personal subjective norms, and (iii) perceived behavioral control (PBC), which is one's ability to mindfully control one's behavior. These three components together directly impact behavior. Individuals may diverge in their smartphone use behavior in at least two analytically distinct ways, i.e., when they use smartphones in the aware mode and when they use it in the unaware mode. According to our results, using smartphones sometimes involves aware and purposeful activities (e.g., reading a book, searching the web) and sometimes involves unaware, unmindful usage. Thus, the TPB may explain an individual's active decisions and preferred choices regarding the utilization of his or her time, which results in the awareness state and thereby negatively influences some QOL features.

To conclude, while an active lifestyle in which smartphones play an important role is likely to improve one's overall level of satisfaction and QOL factors, our research findings indicate a possible domain in which this effect might be harmful when one's smartphone use is performed in parallel with other activities. Therefore, this work increases the need to differentiate between smartphone use modes for the benefit of users and society.

While we believe that this work presents promising avenues for future research and practice, we are also cognizant of its limitations. First, despite an acceptable response rate [58], we are aware of the limitations of utilizing the Mechanical Turk open-source data collection tool, although this approach has been seen to have practical and methodological benefits [59].

Second, although the Aware and Unaware latent variables were constructed from the data, the Reliability of the Aware construct is not high. Although the Aware construct is built from compounds that positively effect QOL, while the Unaware construct negatively effects it, this might suggest that the latent variable itself does not represent one singe usability compound, but rather an accumulated effect of usability variables that contribute to the QOL. One also

should note that the labels, "Aware" and "Unaware" are labels of constructs. Thus, for the Aware construct, it might be that it is a larger set of attribute that have a common property—that they support and improve QOL.

Last, our data included a mid-range number of participants, and it is important to keep in mind that our empirical setting involved individual smartphone users who provided self-report data on the Mechanical Turk online platform. Questions therefore remain about how our results can be generalized to other contexts, such as wider aspects of society and the general population, for which the focus may differ. Additionally, we believe there is a need to complement the existing survey-based measures, which provide mostly static (or at best episodic) snapshots of QOL over an individual's life cycle (see, for example, [60]) and thus may be susceptible to various forms of self-report bias.

## Conclusion

This study examines an unsolved debate regarding the contradictory effects of smartphone use on quality of life. While prior studies have reported mixed and contradicting effects regarding the relationship between smartphone use and QOL, our main contribution to the field is our inspection of the problem through the lens of the mental states of the users while they are utilizing the technology. By separating user behavior into two distinct mental states, namely, aware use and unaware smartphone use, we seem to successfully address the contradiction noted above. Although the use of smartphones in a highly aware state reflects an active lifestyle in which one is likely to feel competent and functioning and to experience positive feelings about one's life, smartphone use in an unaware state is likely to reflect some lack of mindfulness, the absence of concentration, an overload, or similar mental states that can be associated with an addictive and unmindful state of mind. These mental states might also go along higher levels of negative emotions, which the user tries to compensate through an excessive levels of smartphone use and sensory feedback, probably unsuccessfully.

In our digital era, we are all addicted to our smartphones to some degree. Whether this addiction is harmful seems to be mediated by the mental modes in which we use the smartphone. Whether we use a smartphone while in an active concentrated and focused frame of mind or its use while in an absent-minded and split-awareness frame of mind, seems to determine its outcomes to the QOL.

During the COVID-19 pandemic, more than ever before, larger parts of our private and professional lives have been performed through the internet and/or through smartphones. Additionally, the clear separation between our personal and private lives has eroded as people have been forced to work from their homes. Based on our findings, we believe this shift to a remote work-from-home format could potentially worsen the amount of the unaware use of smartphones, since people use smartphones in an ever-growing multi-tasking. We believe that nowadays, more than before, smartphones are not only a gateway that connects individuals to the other individuals in the external world but are also used in performing numerous tasks related to both family and work. As a result, these tasks are now blended in the lives of working-from-home or quarantined workers. Indeed, the worsening of several mental health parameters has been reported and has been associated with excessive smartphone use [60, 61]. This excessive behavior related to unaware smartphone use may result in negative effects, such as stress, fatigue, and addictive online behavior. A better understanding of the smartphone uses modes and their association with QOL regarding remote work scenarios is an important step toward improving one's quality of life while using technology wisely and enhancing a productive and healthy lifestyle.

## Supporting information

**S1 File.**
(DOCX)

## Author Contributions

**Conceptualization:** Alon Sela.

**Formal analysis:** Alon Sela, Noam Rozenboim.

**Funding acquisition:** Hila Chalutz Ben-Gal.

**Investigation:** Alon Sela.

**Methodology:** Alon Sela, Noam Rozenboim, Hila Chalutz Ben-Gal.

**Software:** Noam Rozenboim.

**Supervision:** Alon Sela, Hila Chalutz Ben-Gal.

**Validation:** Alon Sela.

**Visualization:** Alon Sela, Noam Rozenboim.

**Writing – original draft:** Alon Sela, Noam Rozenboim, Hila Chalutz Ben-Gal.

**Writing – review & editing:** Alon Sela, Hila Chalutz Ben-Gal.

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
