## [Decision Letter · Decision Letter 0]

14 Sep 2021

PONE-D-21-20673Smartphone Use Behavior and Quality of Life: What Is the Role of Awareness?PLOS ONE

Dear Author

Thank you for submitting your manuscript to PLOS ONE. After careful consideration, we feel that it has merit but does not fully meet PLOS ONE’s publication criteria as it currently stands. Therefore, we invite you to submit a revised version of the manuscript that addresses the points raised during the review process. Please submit your revised manuscript by 1.10.2021. If you will need more time than this to complete your revisions, please reply to this message or contact the journal office at plosone@plos.org. Please include the following items when submitting your revised manuscript:A rebuttal letter that responds to each point raised by the academic editor and reviewer(s). You should upload this letter as a separate file labeled 'Response to Reviewers'.A marked-up copy of your manuscript that highlights changes made to the original version. You should upload this as a separate file labeled 'Revised Manuscript with Track Changes'.An unmarked version of your revised paper without tracked changes. You should upload this as a separate file labeled 'Manuscript'.

We look forward to receiving your revised manuscript.

Kind regards,

Yuriy Bilan

Academic Editor

PLOS ONE

“This research was partially supported by The Koret Fund of Digital Living 2030.”

“This research was partially supported by The Koret Fund of Digital Living 2030. This research was partially supported by Ariel Cyber Innovation Center.

5. We note that you have referenced (Goldberg, A., Ben-Gal, I., Kreitler S. 2016. “The effect of Smartphone Mobility and Connectivity 641 on Quality of Life”) which has currently not yet been accepted for publication. Please remove this from your References and amend this to state in the body of your manuscript: (Goldberg, A., Ben-Gal, I., Kreitler S. 2016. “The effect of Smartphone Mobility and Connectivity 641 on Quality of Life”. Unpublished Thesis) as detailed online in our guide for authors

http://journals.plos.org/plosone/s/submission-guidelines#loc-reference-style "

Additional Editor Comments (if provided):

Reviewers' comments:

Reviewer's Responses to Questions

**Comments to the Author**

1. Is the manuscript technically sound, and do the data support the conclusions?

Reviewer #1: Partly

Reviewer #2: Yes

2. Has the statistical analysis been performed appropriately and rigorously? 

Reviewer #1: No

Reviewer #2: Yes

3. Have the authors made all data underlying the findings in their manuscript fully available?

Reviewer #1: No

Reviewer #2: Yes

4. Is the manuscript presented in an intelligible fashion and written in standard English?

Reviewer #1: Yes

Reviewer #2: Yes

5. Review Comments to the Author

Reviewer #1: 57) It is worth mentioning that QOL research spans the fields not only of psychology, philosophy, social psychology, but also economics. For example, a noteworthy QOL analysis was performed in DOI: 10.1080 / 1331677X.2021.1956361

143, 179) should improve the description of the methodology for determining the essence of such smartphone usability latent variables as ‘aware’ and ‘unaware’ use behavior modes. In our opinion, it can be learned starting from line 323 of table 1.

185, 195, 219) should improve the description of the methodology for determining the essence of such latent QOL variables as 'competence,' 'functioning,' and 'positive feeling.' In our opinion, it can be found starting from line 325 table 2.

Tables 1 and 2 are mentioned in the text in lines 93, 166, although given in lines 323 and 325.

257) In the Results section, you should leave the information in paragraph 3.4, which clearly corresponds to the title of the article "Smartphone Use Behavior and Quality of Life: What Is the Role of Awareness?"

115) the article does not substantiate the representativeness of the sample of 215 respondents.

323, 325) the name of tables 1 and 2 should be changed, because, as we understand here, the correlation coefficients are given.

323, 325) the names of the graphs in the header are not fully displayed, it is desirable to place the words in a different order so that they can be read by turning the sheet clockwise

291) Figure 3 is mentioned in line 291, and is given in line 316; the signature of the figure is not on the same page with the figure; to make Figure 3 more compact, variable signatures can be matched to Tables 1 and 2: x1-x12, y1-y22.

377) the name of table 3 is detached from the body of the table

368-369) there are errors in the sentence

Reviewer #2: Other benefits of using smartphones and ICT should be considered.

It is necessary to describe how the sample was formed and assess the reliability of the information obtained by the questionnaire.

Some literary sources are outdated.

6. PLOS authors have the option to publish the peer review history of their article (what does this mean?). If published, this will include your full peer review and any attached files.

Reviewer #1: No

Reviewer #2: No

---

## [Author Response · Author response to Decision Letter 0]

31 Oct 2021

The full changes that were made following the reviewer`s comments are found in the "Response to Reviewers" doc file.

---

## [Editor Report · Decision Letter 1]

15 Nov 2021

Smartphone use behavior and quality of life: what Is the role of awareness?

PONE-D-21-20673R1

Dear Authors,

We’re pleased to inform you that your manuscript has been judged scientifically suitable for publication and will be formally accepted for publication once it meets all outstanding technical requirements.

Kind regards,

Yuriy Bilan

Academic Editor

PLOS ONE
---

## [Editor Report · Acceptance letter]

3 Mar 2022

PONE-D-21-20673R1 

Smartphone use behavior and quality of life: what is the role of awareness? 

Dear Dr. Sela:

I'm pleased to inform you that your manuscript has been deemed suitable for publication in PLOS ONE. Congratulations! Your manuscript is now with our production department. 

Kind regards, 

on behalf of

Professor Yuriy Bilan 

Academic Editor

PLOS ONE